# Evaluating the Efficacy of Robot-Assisted Partial Nephrectomy in Complex Renal Tumours: A Single-Centre Retrospective Study

**DOI:** 10.3390/medicina61091702

**Published:** 2025-09-19

**Authors:** Mohammad Hifzi Mohd Hashim, Iqbal Hussain Rizuana, Zulkifli Md Zainuddin, Li Yi Lim, Hau Chun Khoo, Suzliza Shukor, Muhammad Hasif Azizi, Xeng Inn Fam

**Affiliations:** 1Urology Unit, Department of Surgery, Faculty of Medicine, Universiti Kebangsaan Malaysia (UKM), Cheras, Wilayah Persekutuan, Kuala Lumpur 56000, Malaysia; mhifzi91@gmail.com (M.H.M.H.); zuluro@ppukm.ukm.edu.my (Z.M.Z.); limliyi@hotmail.com (L.Y.L.); khoo_daniel85@hotmail.com (H.C.K.); suzliza.shukor@gmail.com (S.S.); hasifazizi@gmail.com (M.H.A.); 2Department of Surgery, Hospital Canselor Tuanku Muhriz, Jalan Yaakob Latif, Bandar Tun Razak, Wilayah Persekutuan, Kuala Lumpur 56000, Malaysia; 3Department of Radiology, Faculty of Medicine, Universiti Kebangsaan Malaysia (UKM), Cheras, Wilayah Persekutuan, Kuala Lumpur 56000, Malaysia; rizi7886@gmail.com; 4Department of Radiology, Hospital Canselor Tuanku Muhriz, Jalan Yaakob Latif, Bandar Tun Razak, Wilayah Persekutuan, Kuala Lumpur 56000, Malaysia; 5Department of Urology, Hospital Picaso, Jalan Professor Khoo Kay Kim, Seksyen 19, Petaling Jaya 46300, Malaysia

**Keywords:** robotic-assisted partial nephrectomy, complex renal tumours, RENAL nephrometry score, perioperative outcomes, Southeast Asia, Trifecta

## Abstract

*Background and Objectives*: Robotic-assisted partial nephrectomy (RAPN) is a preferred minimally invasive option for renal tumours, but its use in highly complex cases (RENAL score ≥ 9) remains underexplored. Only four Asian countries, India, China, South Korea, and Japan, have published studies on RAPN for complex kidney tumours, highlighting limited evidence. The aim of this study is to assess the perioperative, functional, and oncological effects of RAPN for complex renal tumours at a single tertiary centre in Malaysia. *Materials and Methods*: Patient demographics, tumour characteristics, perioperative parameters, and postoperative results were collected through a retrospective review that was conducted on 35 patients who had undergone RAPN between January 2023 and June 2024. The outcomes were analyzed using descriptive statistics, correlation analysis, and comparative tests between surgical approaches (transperitoneal vs. retroperitoneal). *Results*: Of the 35 patients, all had high-complexity tumours. RAPN achieved a “trifecta” outcome in 88.6% of patients. Significantly lower intraoperative blood loss is associated with the retroperitoneal approach in comparison with the transperitoneal approach, whereas other perioperative parameters, which include warm ischaemia time, did not show any significant differences. No positive surgical margins were observed, and no local recurrences or port-site metastases were detected during a mean follow-up of 11.31 ± 5.78 months. Postoperative changes in renal function were negligible, with a mean creatinine change of 5.69 ± 20.39 µmol/L. *Conclusions*: RAPN is a safe and effective option for complex renal tumours, offering excellent functional and oncological outcomes. The choice between transperitoneal and retroperitoneal approaches should be tailored to tumour characteristics for optimal surgical outcomes. This single-centre Malaysian study contributes to the limited Southeast Asian literature on RAPN for complex renal tumours.

## 1. Introduction

Robotic assistance has revolutionized abdominal surgery by addressing the limitations of conventional laparoscopic techniques. Key advantages include enhanced visualization, superior dexterity, and improved precision, which contribute to its growing adoption in urological procedures [1]. Multiple studies have demonstrated that robotic surgeries achieve outcomes comparable, and sometimes superior, to conventional or laparoscopic approaches, with potential benefits such as shorter warm ischemia times and lower complication rates [1,2,3]. Laparoscopic partial nephrectomy (LAPN) is a safe alternative to open partial nephrectomy (PN) with comparable oncological outcomes, better cosmetic and functional outcomes, and shorter hospital stays, but it carries increased risks of operative complications, requires advanced laparoscopic skills, and prolonged warm ischemia during tumour excision and renorrhaphy [4]. Although robotic assistance offers technical advantages for managing complex renal tumours, its broader adoption is often tempered by concerns regarding cost. A study comparing robot-assisted and laparoscopic radical nephrectomy found that the robot-assisted approach increased total direct hospital costs, primarily due to higher operating room expenses [5]. Despite these economic considerations, the benefits of robotic surgery for complex renal tumours, such as improved surgical precision and potentially better perioperative outcomes, warrant further investigation to determine its cost-effectiveness and clinical utility.

There are several nephrometry scoring systems (NSS) that are used to assess renal tumours prior to surgery. Among the NSS are the Preoperative Aspects and Dimensions Used for Anatomical (PADUA) classification; the Radius, Exophytic/Endophytic, Nearness, Anterior/Posterior, Location (RENAL) nephrometry score, centrality index (C-index), diameter-axial-polar (DAP) Nephrometry, and the Arterial-Based Complexity (ABC) scoring systems have been developed to classify renal tumours according to anatomical distributions [6]. The most used classification systems to localize renal carcinomas are the RENAL and PADUA [7]. In this study, we employed the RENAL nephrometry score, which assesses five components: tumour radius (maximum diameter), exophytic/endophytic properties, nearness of the tumour to the collecting system or sinus, anterior/posterior location, and the location relative to polar lines. Higher scores (≥9) indicate increased surgical complexity due to factors such as larger tumour size, central or hilar location, proximity to vital renal structures, or endophytic growth patterns that limit visibility and access during surgery.

Radical nephrectomy has been the standard treatment of localized renal tumours; however, utilization of nephron-sparing surgery or partial nephrectomy was recommended in smaller tumours due to equivalent oncological outcomes with reduced incidence of adverse events in terms of loss of renal function and cardiovascular risk [8]. Robotic-assisted partial nephrectomy (RAPN) is an advanced surgical method that provides a minimally invasive option to traditional open or laparoscopic techniques for the management of renal tumours. RAPN as a minimally invasive surgical treatment for localized renal tumours has been gaining popularity [9]. Its growing adoption is attributed to improved perioperative outcomes and enhanced preservation of renal function [5,10]. While the outcomes of RAPN for localized renal tumours are well-documented, its application in managing complex renal tumours, characterized by a RENAL nephrometry score of 9 or higher, remains relatively unexplored [11,12]. Existing studies on RAPN have exclusively concentrated on the perioperative metrics such as warm ischemia time, predicted blood loss, and complication rates. These studies emphasize the ability of the robotic platform to improve surgical results, primarily in cases that are technically demanding [13,14]. Although RAPN has been widely studied for small or moderately complex renal tumours, its use in highly complex cases (defined by a RENAL nephrometry score of 9 or higher) remains under investigated [15].

Globally, RAPN has been established as a safe and effective approach even for highly complex renal tumours, with Western series reporting perioperative and oncological outcomes comparable to those in less complex cases [9]. In contrast, evidence on RAPN for highly complex renal tumours remains limited in Asia. In a systematic review and meta-analysis of 22 studies, Sharma et al. [16] reported that only four Asian countries, India, China, South Korea, and Japan, have published data, highlighting the scarcity of evidence. This gap is particularly relevant for Southeast Asia, where demographic, clinical, and healthcare system factors may differ from other regions, underscoring the need for region-specific studies to better understand the applicability and outcomes of RAPN [17,18].

This retrospective study evaluates the outcomes of RAPN in managing complex renal tumours (RENAL nephrometry score ≥ 9) at a single tertiary care centre in Malaysia. Warm ischemic time perioperative blood loss, postoperative changes in renal function, length of hospital stay, and surgical margins are the key metrics analyzed to establish the efficacy and safety of this method. The findings aim to contribute valuable insights into the feasibility and effectiveness of RAPN in managing complex renal tumours within this unique regional context. With growing experience in RAPN, its role in managing complex renal tumours should be further explored to justify its potential benefits. More robust data are needed to confirm the feasibility of RAPN as a first-line management option for complex renal tumours, potentially replacing radical nephrectomy.

## 2. Materials and Methods

This study involves a retrospective review of a database that has been prospectively maintained, assessing the results of RAPN conducted by a single surgeon at our institution from January 2023 to June 2024. All patients who underwent RAPN during this period were included, utilizing both transperitoneal and retroperitoneal approaches.

### 2.1. Data Collection

Data collected included patient demographics, tumour characteristics (such as location, size, and complexity), perioperative parameters (including warm ischemia time and blood loss volume), and renal function metrics (both pre- and post-operative serum creatinine and estimated glomerular filtration rate [eGFR]). The difference between the post- and pre-operative values was used for the change in eGFR. Meanwhile, the RENAL nephrometry scoring system was used to assess the tumour complexity, with scores ≥ 9 categorized as highly complex renal tumours. The evaluation of histopathological examination and surgical margin status were also carried out.

### 2.2. Surgical Techniques

All surgeries were performed by a single urologist with extensive experience in minimally invasive and robotic urologic procedures. This consistency helped standardize surgical technique and decision-making across all cases. For this study, both the transperitoneal and retroperitoneal approaches were used for RAPN, depending on the tumour location. The docking of the robotic system and surgical steps were standardized for all cases.

**Transperitoneal Approach**: Patients were positioned in a 70-degree decubitus posture with their flank extended. Pneumoperitoneum was created using a Veress needle, and trocars were placed under direct vision at 8 cm intervals along the semilunar line. The Da Vinci robotic system was docked with targeting. The robotic instruments used consisted of a bipolar grasper, monopolar scissors, ProGrasp forceps, and a large needle driver. Once docking was performed, the colon was medially mobilized to reveal the kidney. The dissection of the renal hilum was performed to isolate the renal artery and vein. Clamping of the artery was then performed using a robotic bulldog clamp, followed by tumour excision using robotic scissors. Barbed sutures were used to repair the renal defect to achieve haemostasis and reconstruct the parenchyma (Figure 1).

**Retroperitoneal Approach**: Patients were positioned in a 90-degree lateral position, and port sites were marked according to the selected approach. A balloon dissector was used to create retroperitoneal space, and trocars were placed at 6 cm intervals using finger guidance or direct vision. Docking of the Da Vinci robotic system was then performed to provide an ideal working space to allow for the robotic arms and camera to obtain accurate surgical access. The bipolar grasper, monopolar scissors, ProGrasp forceps, and a large needle driver were the robotic instruments used for this procedure. The retroperitoneal fat was mobilized, and the lateral coronal fascia was incised. The perirenal fat was then carefully dissected to expose the kidney tumour. The renal hilum was accessed directly, allowing for precise isolation of the renal artery, which was clamped as needed to control blood flow (Figure 2A,B). The tumour was meticulously excised, and the renal defect was repaired in layers using barbed sutures to achieve haemostasis and reconstruct the parenchyma, following the same principles as the transperitoneal approach [19].

Both approaches aim to minimize warm ischemia time (maintained under 30 min wherever possible), ensure complete tumour excision, and achieve secure haemostasis.

### 2.3. Statistical Analysis

SPSS version 26.0 for Windows was used to analyze the data. Descriptive statistics were utilized, with parametric data shown as the mean ± standard deviation and non-parametric data displayed as the median with interquartile range. Normality of continuous variables was assessed using the Shapiro–Wilk test. For normally distributed continuous variables, Pearson’s correlation was used to assess linear relationships between tumour complexity and operative metrics (e.g., warm ischemia time, blood loss). Independent-samples *t*-tests were employed to compare means between the transperitoneal and retroperitoneal surgical approaches, as these groups met assumptions for normality and homogeneity of variances. For categorical variables, Chi-square tests were used to assess associations. A *p*-value of less than 0.05 was considered statistically significant.

### 2.4. Ethics Statement

This study was carried out adhering to the ethical principles outlined in the Declaration of Helsinki (revised 2013). The study protocol was reviewed and approved by the Research Ethics Committee of The National University of Malaysia (IRB no. JEP-2024-804). Written informed consent for surgery was obtained from all patients prior to the procedure. The necessity to obtain additional consent for data publication was waived by the Institutional Review Board of the National University of Malaysia due to the retrospective study design.

### 2.5. Outcomes

The primary outcome of the study was achieving the ‘trifecta’, defined as negative surgical margins, absence of severe perioperative complications (Clavien-Dindo grade ≥ II), and preservation of renal function or a warm ischaemia time of less than 25 min. Secondary outcomes included operative time, warm ischemia time, estimated blood loss, length of hospital stay, and changes in renal function.

## 3. Results

A total of 35 patients, comprising 21 males and 14 females with a mean age of 53.37 ± 15.52 years, underwent RAPN during the study period (Table 1). Tumour characteristics are summarized in Table 2, and perioperative outcomes are detailed in Table 3 and Table 4. All patients had high-complexity tumours, with a median RENAL nephrometry score of 10 (Figure 3A–C).

### 3.1. Correlation Analysis

Higher RENAL nephrometry scores were moderately associated with longer warm ischemia times (correlation coefficient: 0.35) and greater intraoperative blood loss (correlation coefficient: 0.47). A weak positive correlation was observed with length of hospital stay (correlation coefficient: 0.27), while changes in postoperative creatinine levels showed a negligible correlation (correlation coefficient: 0.07). These findings indicate that increased tumour complexity is associated with greater surgical challenges and marginally prolonged recovery times. Creatinine levels remain stable when warm ischemia time (WIT) is less than 20 min, and minimal normal parenchymal tissue is excised, as observed in most of our cases.

### 3.2. Comparative Analysis of Surgical Approaches

An independent-samples *t*-test comparing the transperitoneal and retroperitoneal approaches revealed no statistically significant difference in warm ischaemia time (t = −0.85, *p* = 0.40). However, the retroperitoneal approach was associated with significantly lower intraoperative blood loss compared to the transperitoneal approach (131.0 ± 85.6 mL vs. 242.0 ± 165.3 mL; mean difference −111 mL, 95% CI from −208.7 to −13.3; t = −2.37, *p* = 0.028). Postoperative hospital stay did not differ significantly between retroperitoneal and transperitoneal (2.6 ± 0.7 vs. 2.9 ± 0.9 days; mean difference −0.27 days, 95% CI −0.85 to 0.31; *p* = 0.352). Similarly, no significant difference was observed in postoperative changes in serum creatinine (t = 1.27, *p* = 0.21).

### 3.3. Warm Ischemia Time and Renal Function

The correlation between warm ischaemia time (WIT), and the percentage change in serum creatinine was negligible and negative (r = −0.04, *p* = 0.403). Regression analysis yielded an R^2^ value of 0.0016, indicating that less than 1% of the variance in creatinine changes could be explained by WIT. These findings suggest that WIT has minimal impact on postoperative renal function, particularly when maintained at less than 20 min, as observed in most of our cases. Instead, renal outcomes are more likely influenced by other factors, such as baseline renal reserve. WIT was also comparable between RP and TP (14.3 ± 5.8 min vs. 16.0 ± 5.9 min; mean difference −1.7 min, 95% CI from −5.8 to 2.4; *p* = 0.404).

### 3.4. Operative Console Time and Tumour Complexity

The correlation between the RENAL nephrometry score and operative console time was weakly positive (r = 0.22), suggesting a slight increase in operative console time with higher tumour complexity; however, this relationship was not statistically significant (*p* = 0.206). Similarly, no significant difference in operative console time was observed between the retroperitoneal and transperitoneal approaches (147.7 ± 51.8 min vs. 143.5 ± 35.4 min; mean difference 4.2 min, 95% CI from −25.9 to 34.2; t = 0.28, *p* = 0.779).

### 3.5. “Trifecta” Achievement and Complications

The “trifecta” outcome, absence of severe perioperative complications, negative surgical margins, and preservation of renal function, was achieved in 31 patients (88.6%). There were four cases (11.4%) of Clavien-Dindo Grade ≥ 2 complications (Table 5). These included a postoperative fever requiring intravenous antibiotics, ileus managed conservatively, lung atelectasis treated with non-invasive ventilation, and a transient ischemic attack (TIA) requiring intensive care monitoring. All patients recovered without long-term complications and were discharged within one week.

### 3.6. Histopathological Findings

The histopathological findings reveal a diverse range of tumour types among the patients who underwent robot-assisted partial nephrectomy. Clear cell renal cell carcinoma was the most common, identified in twenty-two patients, followed by angiomyolipoma in seven patients. Additionally, two patients exhibited papillary RCC, and another two had multiloculated cystic RCC. Notably, a rare case of Ewing sarcoma of the kidney was also observed. Importantly, none of the cases showed a positive surgical margin, and no local recurrence or port-site metastases were detected in RCC patients during the mean follow-up period of 11.31 ± 5.78 months.

## 4. Discussion

This study demonstrates that RAPN is a safe and effective surgical option for managing complex renal tumours, defined by a RENAL nephrometry score of ≥9. The favourable perioperative and postoperative outcomes align with findings reported in the literature, emphasizing the role of RAPN in handling high-complexity renal tumours [13,14,20].

The correlation analysis revealed that higher RENAL nephrometry scores were moderately associated with longer warm ischemia times and greater intraoperative blood loss. This finding aligns with the established challenges of managing complex renal tumours, which often require meticulous dissection and reconstruction. Longer ischemic durations are sometimes necessary to balance tumour resection and parenchymal preservation [13,14,20]. The weak correlation with hospital stays duration and the negligible association with serum creatinine changes were expected, as most patients experienced early recovery due to the benefits of minimally invasive surgery. Notably, renal function is significantly affected when WIT is severely prolonged or when a significant amount of normal kidney tissue is resected. These findings highlight that tumour complexity impacts surgical parameters more than immediate recovery or renal function outcomes [21]. Additionally, console time is often prolonged in cases with toxic renal fat, where more time is required to mobilize adhesive perirenal fat. In this study, WIT was consistently kept below 30 min, and there were no cases of severe WIT prolongation, despite tumour complexity. Therefore, console time was not significantly correlated with the RENAL nephrometry score.

Our strategic approach of tailoring the surgical technique to tumour location likely contributed to the favourable outcomes observed. Generally, the retroperitoneal approach was preferred for upper pole, midline, and posteriorly located tumours, while the transperitoneal approach was favoured for lower-pole and anteriorly located tumours. This selection allowed for direct access to the tumour, facilitating optimum excision and suturing. The retroperitoneal approach provides better access to the renal hilum and reduces blood loss compared to the transperitoneal approach. This finding aligns with studies that report reduced bleeding due to optimized visualization in retroperitoneal surgeries [12,14]. Notably, our tumour-tailored approach resulted in comparable warm ischemia times and hospital stays between the two approaches, suggesting that both techniques are viable options depending on tumour characteristics and surgical expertise.

The mean WIT of 15.03 ± 5.84 min in this study is consistent with reported averages for RAPN, typically ranging from 15 to 18 min [19,22]. Importantly, the negligible correlation between WIT and postoperative serum creatinine changes reinforces the notion that renal functional outcomes depend on factors such as baseline renal reserve rather than ischaemic duration alone [13,14]. Furthermore, renal function is well-preserved, particularly when WIT is kept below 20 min [23].

The complete negative surgical margin rate and the absence of local recurrences or port-site metastases during a mean follow-up of 11.31 ± 5.78 months underscore the oncological safety of RAPN for managing complex renal tumours. These findings are consistent with the existing literature, which reports excellent long-term cancer control with RAPN [14,24,25]. For instance, a systematic review by Vartolomei et al. [26] analyzed multiple RAPN series and reported positive surgical margin rates ranging from 0% to 10.5%, with local recurrence occurring in up to 3.6% of patients. Our study achieves excellent results in this parameter, further reinforcing the oncological efficacy of RAPN in complex renal tumours. While no local recurrences or port-site metastases were observed during the follow-up period, it is important to note that our mean follow-up duration is relatively short. As such, the current data do not allow for definitive conclusions regarding long-term oncological control or renal function preservation. Recurrence and delayed functional decline may manifest beyond this period. Therefore, extended follow-up is essential to validate the durability of these outcomes.

The high rate of ‘trifecta’ achievement (88.6%), defined as the absence of severe perioperative complications, negative surgical margins, and preservation of renal function, underscores the efficacy and safety of RAPN in managing complex kidney tumours. The low rate of significant complications further supports its utility, with only 11.4% of patients experiencing Clavien-Dindo Grade ≥ 2 events. These outcomes emphasize the reliability of RAPN for treating complex renal tumours [27]. Our results align with global studies that report similarly high “trifecta” achievement rates for RAPN. Furukawa et al. [28] conducted a large multicentre study in Japan and reported a trifecta rate of 89%, demonstrating the consistency of RAPN outcomes across diverse populations [28]. Notably, our trifecta achievement is comparable with this result, despite exclusively including complex kidney tumours.

Evidence on RAPN for highly complex renal tumours is limited in Asia. Sharma et al. [16] conducted a systematic review and meta-analysis including 22 studies, highlighting that only four Asian countries (India, China, South Korea, and Japan) have published data. For example, a retrospective study from China developed the R.O.A.D score to assess the feasibility of nephron-sparing surgery for hilar tumours [6]. The literature specific to Southeast Asia remains sparse, where demographic, clinical, and healthcare system factors may differ from Western countries [17,18]. This gap underscores the importance of generating regional evidence to contextualize RAPN outcomes.

Our study directly addresses this gap by being the first to report RAPN outcomes for complex renal tumours in Malaysia, thereby contributing valuable data to the Southeast Asian literature. While studies from broader Asian regions, such as China and South Korea, have demonstrated RAPN’s safety and efficacy for complex tumours, comparable regional data have been lacking [20,28,29,30]. A study by Hinata and Fujisawa [31] discussed RAPN techniques and outcomes, highlighting the advantages of the procedure in minimally invasive surgery and nephron-sparing approaches. This study provides valuable insights, but it does not specifically address the Southeast Asian context. Similarly, Pandolfo, Wu, Campi, Bertolo, Amparore, Mari, Verze, Manfredi, Franco, Ditonno, Cerrato, Ferro, Lasorsa, Contieri, Napolitano, Tufano, Lucarelli, Cilio, Perdonà, Siracusano, Autorino, and Aveta [17] conducted a systematic review on RAPN for renal hilar masses, focusing on surgical techniques and outcomes. Although comprehensive, this review lacks data specific to Southeast Asian populations.

The scarcity of region-specific studies underscores the need for more research on RAPN outcomes in Southeast Asia. Factors such as genetic diversity, healthcare infrastructure, and surgical expertise may influence outcomes, making it essential to contextualize findings within the region. According to a study, experienced surgeons for RAPN were linked to lower operative time, WIT, and major postoperative complications [4]. Our findings align with these studies, showcasing RAPN’s ability to achieve favourable perioperative and oncological outcomes in a Southeast Asian context, further validating its potential benefits for this region’s unique demographic and clinical settings.

This study has a number of limitations. Its retrospective nature and small sample size reduce statistical power and limit the generalizability of the findings. Although the outcomes align with the existing global literature, these results should be interpreted with caution. Another limitation is the relatively short follow-up duration (mean 11.3 months), which does not allow for the assessment of long-term oncologic outcomes such as recurrence or metastatic progression. Future larger, multi-centre studies with extended follow-up are required to confirm the robustness and durability of these findings and to provide stronger evidence for clinical practice. Comparative studies between RAPN and alternative surgical approaches could also further delineate its advantages for complex renal tumours.

## 5. Conclusions

This study demonstrates that RAPN is a safe and effective surgical approach for managing complex renal tumours (RENAL nephrometry score ≥ 9), achieving high rates of ‘trifecta’ outcomes. The procedure resulted in positive perioperative and oncological outcomes, such as reduced blood loss, maintained renal function, and lack of positive surgical margins, even in highly complex cases. The choice of approach, whether transperitoneal or retroperitoneal, should be tailored to the characteristics of the tumour to optimize surgical outcomes.

This study represents a significant contribution to the limited data on RAPN outcomes in Southeast Asia, specifically in Malaysia, and aligns with global findings that highlight RAPN as a reliable option for nephron-sparing surgery. Despite being a single-centre study with a modest sample size, this is the first Malaysian study evaluating RAPN for complex renal tumours, and offers important preliminary data. Future larger-scale, multi-centre studies are warranted to validate and expand upon these findings.

## Figures and Tables

**Figure 1 medicina-61-01702-f001:**
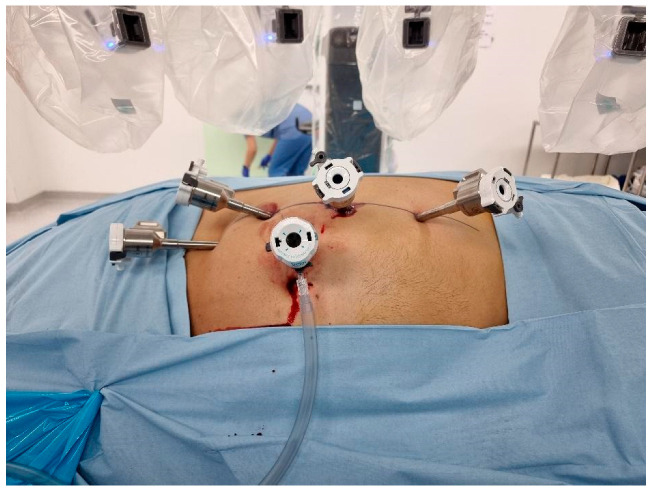
Intraoperative view of transperitoneal robotic port placements. To ensure effective access to the intended site for accurate surgical dissection and reconstruction, the ports were positioned strategically to optimize range of movement of the robotic arms and camera.

**Figure 2 medicina-61-01702-f002:**
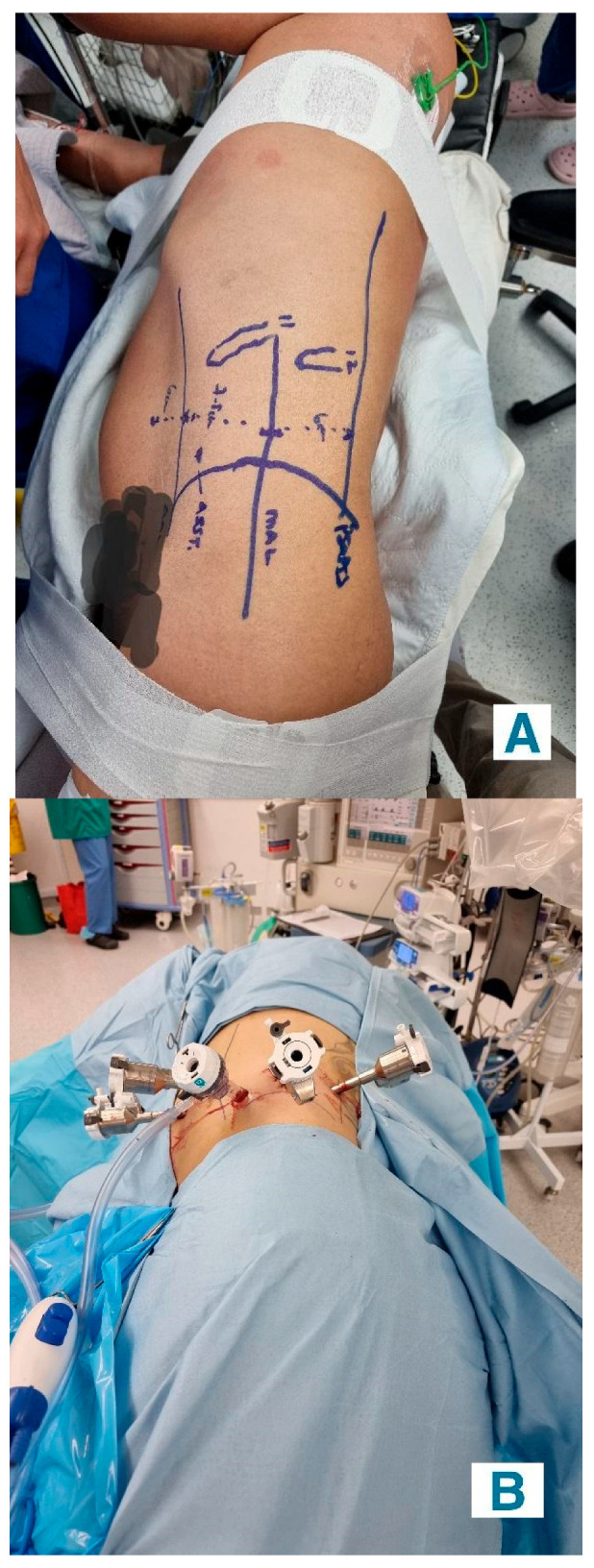
(**A**) Pre-operative marking for retroperitoneal robotic port placements. The markings outline key anatomical landmarks, including the anterior superior iliac spine (ASIS), mid-axillary line (MAL), and the 11th and 12th ribs to guide precise trocar positioning. (**B**) Intraoperative view of retroperitoneal port placements for robot-assisted surgery. The ports are spaced appropriately to ensure optimal triangulation for robotic arm access and facilitate effective instrument maneuverability during the procedure.

**Figure 3 medicina-61-01702-f003:**
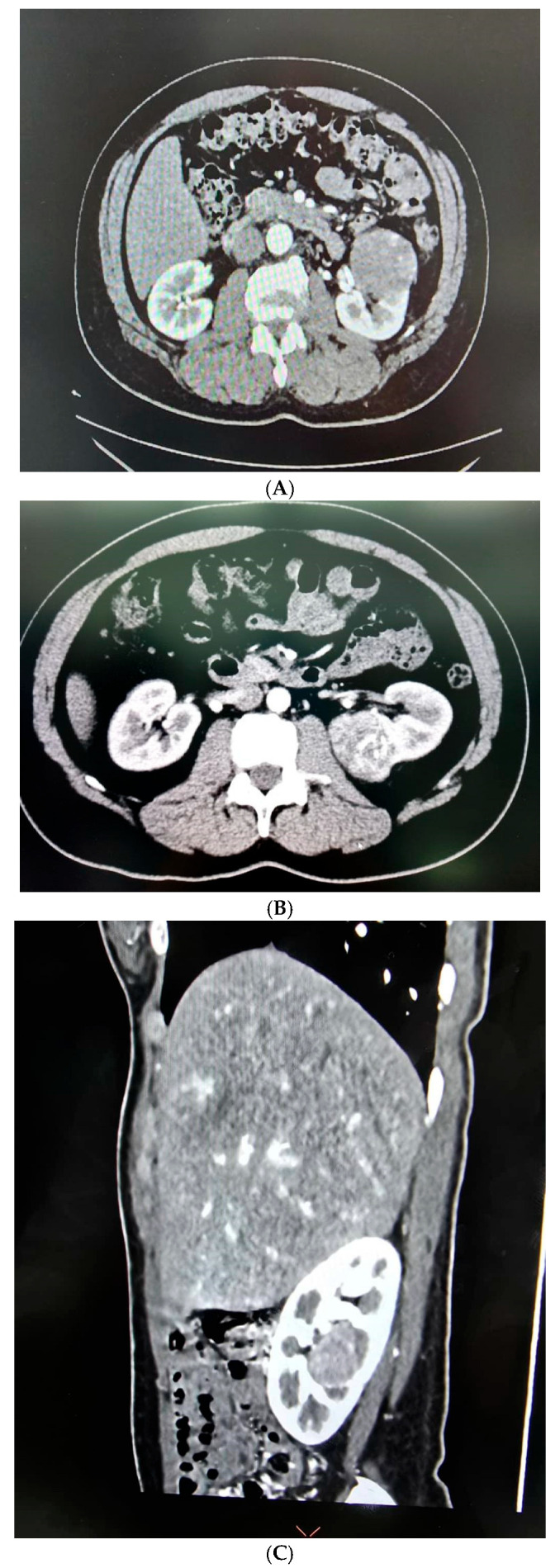
(**A**) Nephrometry score of 10a was given based on the following criteria: (R)adius (2 point); (E)xophytic < 50% (2 points); (N)earness < 4 mm (3 points); (A)nterior; and (L)ocation relative to polar lines (3 points). (**B**) Nephrometry score of 9p was given based on the following criteria: (R)adius (1 point); (E)xophytic < 50% (2 points); (N)earness < 4 mm (3 points); (P)posterior; and (L)ocation relative to polar lines (3 points). (**C**) Nephrometry score of 10× was given based on the following criteria: (R)adius (1 point); (E)ndophytic entirely (3 points); (N)earness < 4 mm (3 points); (X)central; and (L)ocation relative to axial line (3 points).

**Table 1 medicina-61-01702-t001:** Demographic data of patients.

Variables	Number of Patients
Total number of patients (*n*)	35
Age (mean ± SD; range)	53.37 ± 15.52 (22–78) years
Gender (number; percentage)	
Male	21 (60.0%)
Female	14 (40.0%)

**Table 2 medicina-61-01702-t002:** Characteristics of renal lesions and operative approach of the tumour.

Variables	Number of Patients (Percentage)
*Tumour Size*	
<4 cm	13 (37.1)
4–7 cm	19 (54.3)
>7 cm	3 (8.6)
*Kidney side*	
Right	20 (57.1)
Left	15 (42.9)
*Anterior or Posterior*	
Anterior	16 (45.7)
Central	12 (34.3)
Posterior	7 (20.0)
*Tumour Location*	
Upper Pole	6 (17.1)
Midpole	18 (51.4)
Lower Pole	11 (31.4)
*Exophytic and Endophytic*	
Exophytic	19 (54.3)
Partially Endophytic	11 (31.4)
Completely Endophytic	5 (14.3)
*Hilar Tumour*	
Yes	12 (34.3)
No	23 (65.7)
*RENAL Nephrometry Score*	
9	12 (34.3)
10	16 (45.7)
11	6 (17.1)
12	1 (2.9)
*Operative Approach*	
Transperitoneal	15 (42.9)
Retroperitoneal	20 (57.1)

**Table 3 medicina-61-01702-t003:** Perioperative outcomes.

	Mean	Standard Deviation (SD)
Operative Console Time (min)	145.91	±44.96
Warm Ischaemic Time (min)	15.03	±5.84
Blood loss (mL)	178.57	±135.82
Duration of admission (Day)	2.71	±0.79
Creatinine change (µmol/L)	5.69	±20.39

**Table 4 medicina-61-01702-t004:** Oncological and histopathological outcomes.

	Results (Percentage)
*Surgical Margin*	
Positive	0 (0)
Negative	35 (100)
*HPE Findings*	
Clear Cell Renal Cell Carcinoma	22 (62.9)
Papillary Renal Cell Carcinoma (RCC)	2 (5.7)
Multiloculated cystic RCC	2 (5.7)
Angiomyolipoma (AML)	7 (20.0)
Ewing Sarcoma of Kidney	1 (2.9)
Benign Vascular Lesion	1 (2.9)

**Table 5 medicina-61-01702-t005:** Postoperative complications categorized by Clavien-Dindo classification.

Clavien-Dindo Grade	Complication	Number of Patients	Management
Grade II	Postoperative fever	1	IV antibiotics for 7 days
Grade II	Ileus	1	Conservative: bowel rest, IV fluids
Grade II	Lung atelectasis	1	Non-invasive ventilation (NIV), supportive care
Grade II	Transient ischemic attack (TIA)	1	ICU monitoring, resolved without sequelae
Total	-	4 (11.4%)	-

## Data Availability

No datasets were generated or analyzed during the current study.

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
