# Peer review of "Evaluating the Efficacy of Robot-Assisted Partial Nephrectomy in Complex Renal Tumours: A Single-Centre Retrospective Study"

_medicina, 2025, doi:10.3390/medicina61091702_

Round 1
Reviewer 1 Report
Comments and Suggestions for Authors
Dear editors and authors:
It is a great honor and pleasure for me to be invited as the reviewer for this important work entitled “Evaluating the Efficacy of Robotic Assisted Partial Nephrectomy in Complex Renal Tumours: A Single-Centre Retrospective Study” by. Mohammad Hifzi Mohd Hashim and co-authors. This manuscript contributes valuable Southeast Asian data on the use of Robotic-Assisted Partial Nephrectomy (RAPN) for high-complexity renal tumors. It is well-structured and provides a clear rationale, robust methodology, and meaningful results that support the conclusions. The topic is interesting, addressing a gap in the literature in the context of a Malaysian tertiary center.
However, a few areas require clarification, expansion, or revision to strengthen its scientific rigor and improve readability.
Strengths
1.Novel Regional Contribution helps build much-needed literature in Southeast Asia as first known Malaysian study focused specifically on RAPN for complex renal tumors with RENAL nephrometry score ≥ 9.
2.Well-documented inclusion criteria, data collection methods, and statistical tools make the Methodology clear.
3.The retrospective design for this kind of clinical outcome study is appropriate.
- High trifecta rate (88.6%) with excellent safety and functional outcomes provide potential Clinical significance.
- No positive surgical margins or local recurrence is impressive for high-complexity cases.
- Comparing transperitoneal vs retroperitoneal RAPN (TP vs RP) adds valuable nuance, especially the finding of lower blood loss with RP.
Major Comments
- Small Sample Size & Limited Statistical Power: The study only includes 35 patients. While understandable for a single center, this limits generalizability and statistical strength. Authors should acknowledge this limitation more explicitly in the Discussion. Also, consider performing a power analysis (retrospectively) to support the findings' robustness.
- Follow-up Duration: Mean follow-up of ~11 months is insufficient for long-term oncologic assessment. Authors should highlight this in the limitations. I suggest longer follow-up for future prospective studies.
- Trifecta Definition: “Trifecta” is mentioned but not defined in the Methods.
Authors should clearly define the criteria for “trifecta” outcomes — typically margin status, warm ischemia time, and complication rate or eGFR preservation.
- Statistical Clarity: The description of statistical methods lacks specific p-values or confidence intervals in some comparisons. Authors should provide actual p-values and effect sizes (or CI) when comparing RP vs TP.
Minor Comments
1.Language & Style: Overall good, but there are occasional awkward or overly passive phrasings (e.g., "was carried out" instead of "was conducted").
2.Table Presentation: Tables are helpful but consider adding significance indicators (like asterisks) where relevant differences exist.
3.Literature Review: Slightly expand the Introduction with a bit more global context (e.g., RAPN outcomes in Western countries for RENAL ≥9) to better position your study.
4.Ethics & Data Transparency: Ethics approval and retrospective design are clearly stated.
5.English editing with Minor to Moderate Revisions.
This is a valuable manuscript that contributes original clinical data from a region with limited representation in urological surgical outcomes. With minor improvements in methodological transparency and a stronger limitations section, it will be a strong candidate for publication.
Author Response
Comments 1:
Small Sample Size & Limited Statistical Power: The study only includes 35 patients. While understandable for a single center, this limits generalizability and statistical strength. Authors should acknowledge this limitation more explicitly in the Discussion. Also, consider performing a power analysis (retrospectively) to support the findings' robustness.
Response 1:
We thank the reviewer for this important observation. We agree that the relatively small sample size is a limitation and restricts the generalizability of our findings. To address this, we have explicitly acknowledged the issue of limited statistical power in the Discussion (page 12, lines 383–390).
Regarding the reviewer’s suggestion on retrospective power analysis, given the retrospective design and small cohort, a formal post hoc power calculation was not performed, as it may be misleading when applied to non-randomized studies. Instead, we have emphasized in the revised text that the findings should be interpreted with caution and validated in larger, multi-centre studies.
“This study has a number of limitations. Its retrospective nature and small sample size reduce statistical power and limit the generalizability of the findings. Although the outcomes align with existing global literature, these results should be interpreted with caution…
…Future larger, multi-centre studies with extended follow-up are required to confirm the robustness and durability of these findings and to provide stronger evidence for clinical practice.”
Comments 2:
Follow-up Duration: Mean follow-up of ~11 months is insufficient for long-term oncologic assessment. Authors should highlight this in the limitations. I suggest longer follow-up for future prospective studies.
Response 2:
We thank the reviewer for this important suggestion. We agree that the relatively short follow-up period (mean ~11 months) is a limitation, as it may not fully capture long-term oncologic outcomes such as local recurrence, metastasis, or delayed functional decline. Accordingly, we have added this point to the Discussion under the limitations section (page 12, lines 383–390).
The revised passage now reads:
“Another limitation is the relatively short follow-up duration (mean 11.3 months), which does not allow for assessment of long-term oncologic outcomes such as recurrence or metastatic progression. Future larger, multi-centre studies with extended follow-up are required to confirm the robustness and durability of these findings and to provide stronger evidence for clinical practice.”
Comments 3:
Trifecta Definition: “Trifecta” is mentioned but not defined in the Methods. Authors should clearly define the criteria for “trifecta” outcomes — typically margin status, warm ischemia time, and complication rate or eGFR preservation.
Response 3:
We thank the reviewer for pointing this out. We agree that the definition of “trifecta” should be stated clearly in the Methods section. Accordingly, we have revised Section 2.5 (Outcomes) to explicitly define “trifecta” as the absence of severe perioperative complications (Clavien-Dindo grade ≥ II), negative surgical margins, and preservation of renal function. This clarification can be found in Methods (page 6, lines 200–202) of the revised manuscript.
The revised passage now reads:
“The primary outcome of the study was achieving the ‘trifecta,’ defined as negative surgical margins, absence of severe perioperative complications (Clavien-Dindo grade ≥ II), and preservation of renal function or a warm ischaemia time of less than 25 minutes.”
Comments 4:
Statistical Clarity: The description of statistical methods lacks specific p-values or confidence intervals in some comparisons. Authors should provide actual p-values and effect sizes (or CI) when comparing RP vs TP.
Response 4:
We thank the reviewer for this constructive suggestion. We agree that reporting exact p-values and confidence intervals improves clarity and transparency. Accordingly, we have revised the Results section to provide actual p-values and effect sizes (95% CI) for RP vs TP comparisons. These changes are highlighted in the Result segment of the revised manuscript, and are revised as below:
- Estimated blood loss (page 9, lines 245-248)
- Length hospital stay (page 9, lines 248-250)
“However, the retroperitoneal approach was associated with significantly lower intraoperative blood loss compared with the transperitoneal approach (131.0 ± 85.6 mL vs. 242.0 ± 165.3 mL; mean difference −111 mL, 95% CI −208.7 to −13.3; t = −2.37, p = 0.028). Postoperative hospital stay did not differ significantly between retroperitoneal and transperitoneal (2.6 ± 0.7 vs. 2.9 ± 0.9 days; mean difference −0.27 days, 95% CI −0.85 to 0.31; p = 0.352).”
- Warm ischaemic time (page 10, lines 260–262)
“WIT was also comparable between RP and TP (14.3 ± 5.8 min vs. 16.0 ± 5.9 min; mean difference −1.7 min, 95% CI −5.8 to 2.4; p = 0.404).”
- Operative time (page 10, lines 268–270)
“Similarly, no significant difference in operative console time was observed between the retroperitoneal and transperitoneal approaches (147.7 ± 51.8 min vs. 143.5 ± 35.4 min; mean difference 4.2 min, 95% CI −25.9 to 34.2; t = 0.28, p = 0.779).”
Comments 5:
Language & Style: Overall good, but there are occasional awkward or overly passive phrasings (e.g., "was carried out" instead of "was conducted").
Response 5:
We thank the reviewer for this helpful observation. We have carefully revised the manuscript to improve clarity and reduce overly passive constructions. For example, in the Abstract section (page 1, line 28), the phrase “a retrospective review that was carried out” has been revised to “a retrospective review was conducted”. Similar minor adjustments were made throughout the text to enhance readability and style.
Comments 6:
Literature Review: Slightly expand the Introduction with a bit more global context (e.g., RAPN outcomes in Western countries for RENAL ≥9) to better position your study.
Response 6:
We thank the reviewer for this helpful suggestion. We agree that starting with a global perspective provides better flow before focusing on Asian evidence. Accordingly, we have added a brief statement on Western/global outcomes prior to the paragraph on Asian literature. This addition can be found in the Introduction (page 3, lines 97-99), and reads:
“Globally, RAPN has been established as a safe and effective approach even for highly complex renal tumours, with Western series reporting perioperative and oncological outcomes comparable to those in less complex cases [9].”

Reviewer 2 Report
Comments and Suggestions for Authors
The study lack novelty and originality. The authors insisted to highlight the value of a study in local settings. However, the main conclusions are already supported in the existing literature. The discussion part would benefit in more citations of current literature on the topic in the light of study results. The quality of presentation is good. I have only minor corrections to be considered.
Line 41 - 44 – the last two sentences should be deleted.
Line 45 – 47 - should be deleted
Lina 102 -112 – consider incorporate of this part from Introduction section to the Discussion section
Author Response
Comments 1:
The study lacks novelty and originality. The authors insisted to highlight the value of a study in local settings. However, the main conclusions are already supported in the existing literature. The discussion part would benefit from more citations of current literature on the topic in the light of study results.
Response 1:
We sincerely thank the reviewer for this thoughtful comment. We agree that our study conclusions are consistent with international literature, and we acknowledge that the overall novelty is limited. However, we believe that the strength of this study lies in providing the first Malaysian single-centre data on RAPN for complex renal tumours, thereby addressing a current gap in the Southeast Asian context. With the current references, we have ensured that the Discussion highlights how our findings support and complement existing international evidence.
Comments 2:
Line 41–44 – the last two sentences should be deleted.
Response 2:
Thank you for pointing this out. We agree with the reviewer’s suggestion that the abstract should avoid subjective and speculative statements. Therefore, we have revised the last part of the abstract to be more concise and objective. The updated version now reads:
“This single-centre Malaysian study contributes to the limited Southeast Asian literature on RAPN for complex renal tumours.”
This change can be found in the Abstract, page 1, lines 41–42 of the revised manuscript.
Comments 3:
Line 45–47 – should be deleted.
Response 3:
We thank the reviewer for highlighting this. We agree with the suggestion, and the editorial instruction text has been removed.
Comments 4:
Line 102–112 – consider incorporating this part from the Introduction section to the Discussion section.
Response 4:
We thank the reviewer for this valuable suggestion. We agree that the detailed description of existing Asian literature and the evidence gap is more appropriate for the Discussion. Accordingly, we have retained a concise statement in theIntroduction (page 3, lines 99–105):
Evidence on RAPN for highly complex renal tumours remains limited in Asia. In a systematic review and meta-analysis of 22 studies, Sharma et al. [16] reported that only four Asian countries, India, China, South Korea, and Japan, have published data, highlighting the scarcity of evidence. This gap is particularly relevant for Southeast Asia, where demographic, clinical, and healthcare system factors may differ from other regions, underscoring the need for region-specific studies to better understand the applicability and outcomes of RAPN [17,18].
The more detailed content has been relocated to the Discussion (page 12, lines 354–366), with minor rephrasing to improve flow and avoid redundancy. The revised passage reads:
“Evidence on RAPN for highly complex renal tumours is limited in Asia. Sharma et al. [16] conducted a systematic review and meta-analysis including 22 studies, highlighting that only four Asian countries (India, China, South Korea, and Japan) have published data. For example, a retrospective study from China developed the R.O.A.D score to assess the feasibility of nephron-sparing surgery for hilar tumours [6]. Literature specific to Southeast Asia remains sparse, where demographic, clinical, and healthcare system factors may differ from Western countries [17,18]. This gap underscores the importance of generating regional evidence to contextualize RAPN outcomes.
Our study directly addresses this gap by being the first to report RAPN outcomes for complex renal tumours in Malaysia, thereby contributing valuable data to the Southeast Asian literature. While studies from broader Asian regions, such as China and South Korea, have demonstrated RAPN’s safety and efficacy for complex tumours, comparable regional data have been lacking [20,28–30].”

Round 2
Reviewer 1 Report
Comments and Suggestions for Authors
After the appropriate English editing, I endorse the publication.